# Fcα Receptor-1-Activated Monocytes Promote B Lymphocyte Migration and IgA Isotype Switching

**DOI:** 10.3390/ijms231911132

**Published:** 2022-09-22

**Authors:** Amélie V. Bos, Melissa M. J. van Gool, Annelot C. Breedveld, Richard van der Mast, Casper Marsman, Gerd Bouma, Mark A. van de Wiel, S. Marieke van Ham, Reina E. Mebius, Marjolein van Egmond

**Affiliations:** 1Department of Molecular Cell Biology and Immunology, Amsterdam UMC Location Vrije Universiteit Amsterdam, Boelelaan 1117, 1081 Amsterdam, The Netherlands; 2Research Institute of Amsterdam Institute for Infection and Immunity, Inflammatory Diseases, 1105 Amsterdam, The Netherlands; 3Sanquin Research and Landsteiner Laboratory, Department of Immunopathology, Amsterdam UMC, 1081 Amsterdam, The Netherlands; 4Department of Gastroenterology and Hepatology, Amsterdam UMC Location Vrije Universiteit Amsterdam, Boelelaan 1117, 1081 Amsterdam, The Netherlands; 5Epidemiology & Biostatistics, Amsterdam UMC Location Vrije Universiteit Amsterdam, Boelelaan 1117, 1081 Amsterdam, The Netherlands; 6MRC Biostatistics Unit, Cambridge University, Cambridge CB2 1TN, UK; 7Swammerdam Institute for Life Sciences, University of Amsterdam, 1012 Amsterdam, The Netherlands; 8Department of Surgery, Amsterdam UMC Location Vrije Universiteit Amsterdam, Boelelaan 1117, 1081 Amsterdam, The Netherlands

**Keywords:** immunoglobulin A, IgA, CD89, FcαRI, myeloid cells, monocytes, inflammatory bowel disease, IBD, CCL20

## Abstract

Patients with inflammatory bowel disease (IBD) produce enhanced immunoglobulin A (IgA) against the microbiota compared to healthy individuals, which has been correlated with disease severity. Since IgA complexes can potently activate myeloid cells via the IgA receptor FcαRI (CD89), excessive IgA production may contribute to IBD pathology. However, the cellular mechanisms that contribute to dysregulated IgA production in IBD are poorly understood. Here, we demonstrate that intestinal FcαRI-expressing myeloid cells (i.e., monocytes and neutrophils) are in close contact with B lymphocytes in the lamina propria of IBD patients. Furthermore, stimulation of FcαRI-on monocytes triggered production of cytokines and chemokines that regulate B-cell differentiation and migration, including interleukin-6 (IL6), interleukin-10 (IL10), tumour necrosis factor-α (TNFα), a proliferation-inducing ligand (APRIL), and chemokine ligand-20 (CCL20). In vitro, these cytokines promoted IgA isotype switching in human B cells. Moreover, when naïve B lymphocytes were cultured in vitro in the presence of FcαRI-stimulated monocytes, enhanced IgA isotype switching was observed compared to B cells that were cultured with non-stimulated monocytes. Taken together, FcαRI-activated monocytes produced a cocktail of cytokines, as well as chemokines, that stimulated IgA switching in B cells, and close contact between B cells and myeloid cells was observed in the colons of IBD patients. As such, we hypothesize that, in IBD, IgA complexes activate myeloid cells, which in turn can result in excessive IgA production, likely contributing to disease pathology. Interrupting this loop may, therefore, represent a novel therapeutic strategy.

## 1. Introduction

IgA is predominantly produced in mucosal tissues (up to 60 mg/kg body weight/day) by local B lymphocytes in the lamina propria [1,2]. Mucosal IgA antibody production is initiated when B lymphocytes recognize the antigen via their B-cell receptor. Consequently, activated B lymphocytes are attracted by chemokines, including CCL20, which are secreted in organized tissues such as Peyer’s patches, where B cells further differentiate with help of other cells and molecules of the immune system [3,4]. In mucosal germinal centers, activated T-helper cells provide CD40-ligand (CD40L) co-stimulation and secrete cytokines to facilitate B-cell differentiation. Additionally, other cytokines have been described to promote IgA differentiation and B cell proliferation, such as a proliferation-inducing ligand (APRIL) [5,6], interleukin-6 (IL6) [7,8], interleukin-10 (IL10) [9], and tumor necrosis factor-α (TNFα) [10]. As such, the extent to which these cytokines are produced regulate and balance IgA differentiation of B lymphocytes in the mucosa.

In humans, IgA antibodies are produced in different forms, which differ in structure and immune-mediated characteristics. While systemic serum IgA is produced in a monomeric structure, mucosal IgA consists of two IgA molecules, creating a dimer with an additional J chain (dimeric IgA; dIgA). In the gut, dIgA binds to the polymeric immunoglobulin receptor (pIgR) that is expressed on epithelial cells, which facilitates transport through the epithelial barrier and secretion into the intestinal lumen as secretory IgA (SIgA). A part of the pIgR remains attached (referred to as secretory component), which partly blocks the IgA Fc-tail [11,12]. SIgA in the lumen maintains a healthy diversified microbiome through its neutralizing properties, prevents microbial invasion, and favors the growth of specific commensal bacteria, such as Firmicutes and Lachnospiracae [13,14,15,16,17,18,19,20]. As a result, SIgA is key player in promoting host–microbiota symbiosis and maintaining homeostasis.

In contrast to SIgA, the Fc-tail of serum IgA and dIgA is freely accessible to bind to its Fcα-receptor I (FcαRI), which is exclusively expressed on myeloid immune cells such as neutrophils, monocytes, eosinophils, macrophages, and subsets of dendritic cells [21,22,23,24,25]. Consequently, SIgA poorly initiates myeloid cell activation, whereas serum IgA and dIgA complexes can crosslink FcαRI and potently induce proinflammatory functions [23,26]. Although monomeric IgA induces inhibitory signals, crosslinking FcαRI by IgA complexes has been described to induce degranulation, phagocytosis, chemotaxis, and antibody-dependent cellular cytotoxicity [27,28,29]. Moreover, serum IgA-mediated myeloid cell activation triggers production of proinflammatory cytokines including interleukin-6 (IL6), tumor necrosis factor-α (TNFα), and interleukin-1β [25]. When myeloid cells are additionally stimulated via Toll-like receptors, the ligands of which are expressed on bacteria and viruses, production of these cytokines is further enhanced [25]. Similarly, dIgA crosslinking resulted in high TNFα production by macrophages, comparable to serum IgA stimulation [25]. In case of an invasive mucosal pathogen, tissue-derived dIgA and Toll-like receptor activation may, therefore, synergize production of proinflammatory cytokines and can be important to prevent and/or resolve an infection.

Inflammatory bowel disease (IBD) manifests as a chronic inflammation of the gastrointestinal tract, and it can be subdivided into Crohn’s disease (CD) and ulcerative colitis (UC). Although its etiology is incompletely understood, it has been proposed that IBD is characterized by excessive immune responses against commensal intestinal bacteria [30,31]. Generally, inflamed sites in UC feature the infiltration of neutrophils [32], whereas CD lesions additionally are heavily infiltrated with mononuclear cells such as monocytes and macrophages [33,34,35,36]. It was observed that neutrophils in the intestinal lamina propria tissue of patients with UC contain IgA [32], suggesting uptake of IgA complexes in situ. Moreover, increased proportions of IgA-coated bacteria have been observed in the feces of IBD patients compared to healthy controls [37,38], and the percentage of IgA-coated bacteria was associated with disease severity. Injection of highly IgA-coated fecal bacteria of IBD patients into germ-free mice led to exacerbated dextran sulfate sodium (DSS)-induced colitis, suggesting that high-level IgA coating identified colitogenic bacteria in IBD [37]. Moreover, a B-cell subset producing high levels of mucosal IgA was shown to exacerbate DSS-induced colitis [39]. Collectively, these data suggest that IgA activation of myeloid cells via FcαRI contributes to IBD pathology. However, the underlying mechanism leading to aberrant and excessive IgA production by B lymphocytes in IBD is poorly understood.

In this study, we investigated the contribution of FcαRI-stimulated myeloid cells to B lymphocyte functionality, including IgA differentiation, which may play a role in enhanced IgA coating of the microbiome as observed in IBD patients.

## 2. Results

### 2.1. Myeloid Cells Interact with B Lymphocytes in Colon Tissue of Patients with Inflammatory Bowel Disease

To study whether mucosal B lymphocytes and myeloid cells are involved in IBD pathogenesis, the infiltration of these subsets was determined in colon biopsies. As such, colon biopsies of macroscopic pathological lesions and adjacent noninflamed areas as an internal control were obtained for microscopic analysis. Inflamed colon tissue of IBD patients with active disease had disrupted tissue integrity and demonstrated a trend toward increased numbers of CD14^+^ and CD66b^+^ cells compared to paired noninflamed tissues (Figure 1A,B), which is consistent with earlier observations [32,33,34,35,36]. In noninflamed tissues, myeloid cells were evenly distributed. Additionally, large clusters of CD66b^+^ neutrophils were observed in inflamed biopsies (Figure 1D). Single CD19^+^ B lymphocytes were evenly distributed throughout all biopsies, and accumulated B-cell clusters were found in both inflamed and noninflamed colon tissues (Figure 1C,D). There was a trend toward increased numbers of CD19^+^ cells in inflamed biopsies of patients compared to noninflamed tissues (Figure 1C) (*p*-value = 0.06). The interaction between neutrophils and IgM^+^ (*R^2^* = 0.33) or IgA^+^ (*R^2^* = 0.60) B cells correlated with the overall numbers of neutrophils in the tissue (Appendix A). The interaction of B cells with CD14^+^ monocytes and CD66b^+^ neutrophils was observed in the lamina propria of inflamed (Figure 1E–H) and paired noninflamed colon tissue of both CD and UC patients (Appendix A).

### 2.2. FcαRI-Stimulated Monocytes Produce the Chemokine CCL20 and Attract B Lymphocytes

Close interaction between myeloid cells and B lymphocytes was observed in the inflamed colon tissue of IBD patients, which led to the postulation that myeloid cells might influence B-cell functions. To mimic activation via IgA complexes, we used IgA-opsonized beads to exclude activation via TLRs. Moreover, mucosal dIgA is difficult to purify because it may become contaminated with microbial components and TLR stimulants, which could interfere with FcαRI myeloid cell activation, due to crosstalk between FcαRI and TLR. We, therefore, used purified pooled serum IgA, as, similarly to dIgA, it potently crosslinks FcαRI on human myeloid cells [40]. Monocytes were stimulated with IgA-opsonized beads, after which RNA sequencing was performed. In total, 1281 genes were significantly differently expressed between FcαRI-stimulated or bovine serum albumin (BSA)-stimulated human peripheral CD14^+^ monocytes (Figure 2A). To gain insight into the potential involvement of FcαRI-stimulated monocytes in B lymphocyte migration, we assessed expression of genes that are known to be important in B-cell chemotaxis. Five chemokines were detected in the dataset, of which C–C motif chemokine ligand 20 (CCL20) was significantly upregulated in FcαRI-stimulated monocytes, compared to monocytes stimulated with BSA-coated beads (Figure 2B). Gene expression of B-cell chemokines CCL28, CCCL19, C–X–C motif chemokine ligand 12 (CXCL12), and CXCL13 was not significantly different after stimulation with BSA- or IgA-coated beads. In line with our RNA sequencing data, we observed that CCL20 protein release was enhanced by FcαRI-stimulated monocytes, whereas CXCL13 protein levels were unaffected (Figure 2C and Appendix A). Next, B-cell migration was assessed. The number of B cells migrating toward the supernatant of monocytes treated with BSA-coated beads was similar compared to the number of cells migrating toward a medium control (Figure 2D,E). In contrast, supernatant of FcαRI-stimulated monocytes triggered enhanced migration of B lymphocytes, supporting that FcαRI-stimulation of monocytes resulted in secretion of B-cell chemokines (Figure 2D,E).

### 2.3. FcαRI-Stimulated Monocytes Produce Interleukin-6, Interleukin-10, and Tumor Necrosis Factor-α

To study whether FcαRI-stimulated monocytes may influence B-cell differentiation, genes known to be involved in B-cell maturation/activation were shortlisted from the RNA sequencing dataset. Six genes were significantly upregulated in FcαRI-activated monocytes, i.e., IL6, IL7, IL10, tumor necrosis factor-α-induced proteins (TNFAIP), and IL27 (Figure 3A). Corresponding to the RNA sequencing dataset, FcαRI-stimulated monocytes secreted enhanced IL6, TNFα, and IL10 protein levels (Figure 3B–D), compared to monocytes cultured with BSA-coated beads. Although APRIL gene expression was unaltered between stimulating monocytes with IgA- compared to BSA-coated beads, we observed that APRIL protein secretion was slightly increased when monocytes were stimulated via FcαRI (*p*-value = 0.06) (Figure 3E). FcαRI-stimulated monocytes did not trigger protein production of B-cell-activating factor (BAFF) (Figure 3F). Moreover, IL7 and IL4 protein secretion in supernatants was below the detection limit (Figure 3G,H). Supplementation of recombinant IL6 or APRIL protein to naïve B lymphocyte cultures induced an increase in the expression of surface IgA, while IL10 and TNFα resulted in a trend of increase (Figure 4). Together, these data demonstrate that FcαRI-stimulated monocytes produced the cytokines IL6, IL10, TNFα, and APRIL, which in turn can induce IgA class switch recombination in naïve human B cells in vitro.

### 2.4. FcαRI-Stimulated Monocytes Induce IgA Class Switching in Naïve B Lymphocytes

Next, we studied whether FcαRI-stimulated monocytes can directly steer IgA class switch recombination in coculture with naïve B cells. The effect of FcαRI-stimulated monocytes on B cells varied per donor and pointed to different levels of B-cell differentiation into antibody-secreting cells after class switching. Some cocultured B cells demonstrated increased IgA membrane expression (Figure 5A,B; d1/d2/d3/d4), whereas additional cultures demonstrated enhanced IgA secretion, while IgA detection was no longer present on the B cells. The effect of FcαRI-stimulated monocytes on B cells varied per donor, as some demonstrated increased IgA membrane expression (Figure 5A,B; d1/d2/d3/d4), whereas others only demonstrated enhanced IgA secretion (Figure 5C,D; d5/d6). Moreover, B cells of donors 2 and 3 had both increased IgA membrane expression and enhanced IgA secretion (d2/d3). B-cell cultures without monocytes but with IgA-opsonized beads served as control to exclude that detected IgA in the supernatants of cocultures was derived from the IgA-opsonized beads used during stimulation of monocytes (Figure 5C and Appendix A). Similarly, we tested whether cocultures of naïve B cells with FcαRI-stimulated human neutrophils also trigger IgA class switch recombination in vitro to support our findings. However, coculturing neutrophils led to naïve B cell death, whereas monocytes had a beneficial effect on naïve B cell survival (Appendix A). Taken together, these data demonstrate that FcαRI-stimulated monocytes are responsible for promoting IgA differentiation of human B cells.

## 3. Discussion

It has been demonstrated that IBD patients produce enhanced IgA against the microbiota. However, the mechanisms that trigger enhanced IgA and its role in IBD disease pathology are incompletely understood. In this study, we demonstrate that FcαRI-stimulated monocytes produce the chemokine CCL20 and recruit B lymphocytes. Moreover, FcαRI-stimulated monocytes secreted the proinflammatory cytokines IL6 and TNFα. This is in line with our recent study demonstrating that monocytes produce a range of proinflammatory cytokines including IL6 and TNFα [41]. IL6 and TNFα have been linked to IBD pathology [42]. IL6/TNFα RNA transcripts [43,44] and protein secretion [45,46,47] were increased in monocytes residing in the lamina propria of IBD patients, which correlated to active disease and tissue damage. Moreover, anti-TNFα antibody therapy is used to treat IBD, as it decreases clinical symptoms in both UC and CD patients [48], providing further evidence for the pathological role of TNFα in IBD pathogenesis.

The levels of soluble and microbiota-specific IgA antibodies have been linked to disease pathology [38]. Previously, a murine study suggested that neutrophils negatively regulate mucosal IgA secretion, as it was observed that sublingual immunization in mice resulted in enhanced specific IgA secretion when neutrophils were depleted [49]. However, mice do not express a natural homolog for FcαRI; therefore, in vivo studies generally exclude the influence of FcαRI-mediated activation in the experimental readouts. By contrast, crosslinking of FcαRI on human neutrophils triggers potent pro-inflammatory functions including degranulation, phagocytosis, chemotaxis, antibody-dependent cellular cytotoxicity [27,28,29], and production of proinflammatory cytokines including IL6, TNFα, and IL1β [25]. This would not occur with murine neutrophils, as they do not express FcαRI. To study the interaction between neutrophil FcαRI and IgA, we developed a transgenic/knock-in mouse model that expresses both human FcαRI and human IgA. DSS-induced colitis in these mice ameliorated disease compared to control animals, suggesting that the interaction between IgA and FcαRI contributes to gut inflammation pathology [41]. The cellular mechanisms underlying enhanced IgA production in IBD are unknown. Here, we observed that culturing human B lymphocytes with recombinant IL6 and TNFα promoted IgA class switching. The capacity of IL6 [7,8], APRIL [5,6], and IL10 [9] to induce IgA B-cell differentiation was also demonstrated by others. Moreover, we demonstrate that coculturing human B lymphocytes with FcαRI-stimulated monocytes, producing these cytokines, resulted in enhanced IgA B-cell membrane expression and secretion. As such, FcαRI-stimulated myeloid cells can positively regulate IgA production in vitro.

Additionally, FcαRI-stimulated monocytes also secreted the anti-inflammatory cytokine IL10. IL10 is known for suppressing proinflammatory cytokines produced by monocytes, such as TNFα and IL6 [50,51,52], and attenuating an activated immune system [52]. During health, it is likely that IL6/ TNFα and IL10 are produced simultaneously to balance pro- and anti-inflammatory responses. However, a polymorphism, leading to a premature stop codon in the IL10 receptor, has been identified in a fraction of IBD patients. As such, it was postulated that they may be less responsive to the dampening effect of IL10 [53,54]. This is reflected in systemic cytokines in IBD patients, as increased TNFα and IL6 production is observed [55,56,57].

During active IBD, intestinal CCL20 is increased [58,59]. A role for CCL20 in IBD pathology was suggested on the basis of genome association and gene expression studies [60,61]. CCL20 may contribute to pathology by promoting recruitment of interleukin-17-producing T cells [62]. In line, experimental colitis in mice was attenuated by treatment with anti-CCL20 neutralizing antibodies, which reduced mucosal T-cell infiltration. In addition, CCL20 facilitates B-cell migration into the Peyer’s patches and isolated lymphoid follicles, contributing to production of mucosal IgA [3,63]. Yet, little is known about CCL20 and B cells in IBD pathology. Although it has been shown that the lack of CCL20-dependent mucosal B cell migration led to reduced IgA production and an altered intestinal flora [3], it is not clear what the effect of excessive CCL20 production is on B cells and IBD pathology. In this study, we observed that FcαRI-stimulated monocytes secreted CCL20 and recruited human B lymphocytes in vitro. As such, it is possible that, during IBD, activated myeloid cells contribute to the recruitment of B cells, which in turn produce increased IgA against the microbiota.

The involvement of IgA in inflammatory disease pathology is presumably not exclusive to IBD. IgA autoantibodies, increased IgA, or aberrant IgA immune complexes are found in several diseases, including celiac disease [64,65], IgA nephropathy [66,67,68], multiple sclerosis [69], IgA blistering diseases [70], and linear IgA bullous disease (LABD) [71]. The pathological contribution of auto-IgA antibodies was recently demonstrated in an LABD in vivo model. It was demonstrated that presence of IgA anti-collagen 17 in the skin of human FcαRI transgenic mice led to neutrophil accumulation and inflammation-mediated tissue damage [72]. In rheumatoid arthritis, it was shown that synovial fluid contains infiltration of neutrophils as well as auto-IgA complexes, which activated neutrophils in vitro [73]. Consequently, a role for FcαRI-activated neutrophils and osteoclasts in inducing joint damage and exacerbation of disease is highly likely, which is supported by the finding that the presence of auto-IgA complexes correlated with disease progression and severity [74,75,76]. It was suggested that IgA autoantibodies induce release of IL6 and IL8 by immune cells, as well as osteoclasts, which enhances bone resorption by osteoclasts, possibly reflecting enhanced rheumatoid arthritis pathogenesis [77]. Similarly, IgA vasculitis is characterized by IgA1 deposits, neutrophil infiltration and vascular inflammation. It was demonstrated that patients with IgA vasculitis produce IgA autoantibodies that can bind to blood vessels, and it has been hypothesized that IgA anti-endothelial autoantibodies may activate neutrophils in an FcαRI-dependent manner and contribute to disease pathology [78].

On the basis of our findings, we propose the following model (Figure 6): during mucosal infection, IgA-opsonized bacteria are recognized by myeloid cells and cleared. B-cell activation factors may be temporarily increased; however, once the infection is cleared, the process stops, and the system returns to homeostasis. However, during IBD, enhanced IgA is produced against the microbiota compared to healthy individuals, possibly because of a damaged epithelial layer. IgA complexes, such as IgA-opsonized bacteria, activate myeloid cells via FcαRI, which produce B-cell chemokines and B-cell-activating factors. As a result, B lymphocytes are recruited to the site of inflammation via CCL20 and are skewed into an IgA isotype. Likely, this mechanism promotes even more IgA production, which in turn can activate newly recruited myeloid cells. Consequently, a continuous positive feedback loop persists that activates myeloid cells and promotes IgA B-cell differentiation.

Taken together, this study demonstrates that FcαRI-stimulated monocytes recruit B cells and promote IgA differentiation. As such, we anticipate that FcαRI-stimulated myeloid cells such as monocytes and neutrophils contribute to IBD pathology. Blocking FcαRI with monoclonal antibodies was shown to decrease neutrophil recruitment and tissue damage in an IgA skin blistering model [72]. Thus, FcαRI blocking therapy may also represent an interesting novel therapeutic strategy for IBD patients.

## 4. Materials and Methods

**Human donor and patient samples.** Colon biopsies were collected from patients undergoing colonoscopy for diagnostic screening. The Medical Ethical Committee of the VU University Medical Center (the Netherlands) evaluated the project (2013/148), and all participants signed informed consent in accordance with their guidelines.

**PMN and monocyte isolation.** Peripheral blood mononuclear cells (PBMCs), polymorphonuclear cells (PMNs), and monocytes were isolated from the peripheral blood of healthy donors and buffy coats (Sanquin blood bank, Amsterdam, the Netherlands). Polymorphonuclear cells (PMNs) were isolated with Lymphoprep (Axis-Shield cat: 1114547, Dundee, UK) density gradient centrifugation, after which erythrocytes were lysed in ammonium chloride buffer (10 min RT, 155 mM NH_4_Cl, 10 mM KHCO_3_, and 0.11 mM EDTA). After lysis, PMNs were washed with PBS and resuspended in RPMI 1640 that was supplemented with 10% FCS, 1% penicillin/streptomycin/glutamine (PSG) (1 h, 37 °C) to rest before continuing with phagocytosis assays. CD14^+^ monocytes were isolated from the PBMC fraction with positive selection using CD14 magnetic microbeads (Miltenyi Biotec, Bergisch Gladbach, Germany) according to the manufacturer’s protocol. Monocytes were kept in PBS (30 min, 4 °C) to rest before continuing with phagocytosis assays.

**Preparation of BSA- and IgA-coated latex beads.** Nonfluorescent 0.9 µm latex beads (Sigma-Aldrich cat: CLB9-1ml, St. Louis, MO, USA) were washed twice with 2-(*N*-morpholino) ethanesulfonic (MES) buffer (30 mM, pH 6.1) and resuspended in MES buffer with 2 mg/mL BSA (Roche Diagnostics, Basel, Switzerland) or serum IgA (MP Biomedicals cat: 0855906, Santa Ana, CA, USA) in the presence of *N*-(3-dimethylaminopropyl)-*N*’-acid ethylcarbodimide hydrochloride (Sigma-Aldrich) and incubated o/n (overhead shaker, 4 °C). Latex beads were washed and resuspended in PBS containing 0.1% BSA.

**B-cell culture.** PBMCs were isolated with Lymphoprep (Axis-Shield) density gradient centrifugation. Naïve B lymphocytes were purified from the PBMC fraction using negative isolation via EasySep™ Human Naïve B-Cell Isolation Kits (Stemcell Technologies, Vancouver, BC, Canada). Naïve B cells were cultured in RPMI (Gibco cat: 11875-093, Waltham, MA, USA) supplemented with 10% FCS, 1% PSG (both Thermo Scientific, Waltham, MA, USA), 1 µg/mL anti-IgM (Clone DA4-4, MyBiosource, San Diego, CA, USA), 20 ng/mL IL-4 (Immunotools, Friesoythe, Germany), and 1 µg/mL anti-CD40 antibody (Clone G28.5 BioXcell, Lebanon, NH, USA) for 7 days.

**Supplementation recombinant cytokines.** Recombinant human 50 ng/mL IL6 (Immunotools cat: 11340066), 10 ng/mL IL10 (BD Biosciences cat: 554611, San Jose, CA, USA), 10 ng/mL TNFα (Peprotech, cat: 300-01A, Waltham, MA USA), and 200 ng/mL APRIL (R&D Systems cat: 5860-AP-010/CF, Minneapolis, MN, USA) were supplemented on day 0 when indicated.

**Monocyte coculture.** CD14^+^ monocytes were stimulated with BSA- and IgA-coated beads at a 1:100 ratio of cells to beads (30 min, 37 °C) in RPMI supplemented with 10% FCS, 1% PSG. Noninternalized beads were washed away with RPMI medium (1500 rpm), and 5 × 10^4^ monocytes were seeded per condition in a U-shaped 96-well plate (Greiner Bio-One, Kremsmünster, Austria). Naïve B cells were labeled with CellTrace™ Violet Cell Proliferation Kit (ThermoFisher cat: 15579992, Waltham, MA, USA) according to the manufacturer’s protocol. A total of 5 × 10^4^ labeled naïve B cells were cocultured with stimulated monocytes (E:T ratio 1:1) After 7 days, cells were harvested and analyzed.

**B-cell migration assay.** CD14^+^ monocytes were stimulated with BSA- or IgA-coated beads at 1:100 cells to beads ratio (overnight, 37 °C) in RPMI supplemented with 10% FCS, 1% PSG. Cells were removed by centrifugation (twice at 1500 rpm), and 120 µL of supernatant was pipetted into the lower compartment of a 5.0 µm pore polycarbonate 96-transwell system (Corning cat: 3388, Corning, NY, USA). Total CD19^+^ B cells were isolated from PBMCs using human Pan-B Cell Enrichment Kit (Stemcell Technologies, cat: 19514, Vancouver, BC, Canada). B cells were labeled with CellTrace™ Violet Cell Proliferation Kit (ThermoFisher cat: 15579992) according to the manufacturer’s protocol. Briefly, 75 µL of medium containing 1 × 10^5^ B cells was pipetted in upper chambers of transwells (3 h, 37 °C). Migrated cells were taken from lower chambers, and Cell Trace-positive events were counted using an LSR-Fortessa X20 (BD Bioscience, San Jose, CA, USA).

**RNA isolation.** Total RNA from human peripheral CD14^+^ monocytes was isolated using TRIzol Reagent (Invitrogen, Waltham, MA, USA) following the manufacturer’s protocol. In brief, human peripheral CD14^+^ monocytes (2 × 10^6^) were lysed in TRIzol Reagent, and chloroform was added to permit complete dissociation of nucleoprotein complexes. Samples were mixed vigorously and centrifuged at 12,000× *g* for 15 min at 4 °C to separate the biphasic mixtures into a lower red, phenol–chloroform phase and an upper colorless, aqueous phase. RNA was precipitated from the aqueous phase with isopropanol and centrifuged at 12,000× *g* for 10 min at 4 °C. The RNA pellet was washed twice with 75% ethanol, air-dried, and resuspended in 30 μL of RNAse/DNase-free water. RNA quality and yield were analyzed with the Agilent RNA 6000 Nano Kit (Agilent Technologies, Santa Clara, CA, USA) according to the manufacturer’s protocol using capillary electrophoreses (Agilent 2100 Bioanalyzer, Agilent Technologies, Santa Clara, CA, USA). RNA samples with an RNA integrity number (RIN) below 7.5 were excluded.

**Real-Time PCR.** Reverse transcription of RNA was performed using the Promega A3500 kit according to the manufacturer’s instructions (Promega, Madison, WI, USA). Melt curves were recorded and analyzed using a StepOne Real-Time PCR System (ThermoFisher, Waltham, MA, USA). Genes of interest (Appendix A) were normalized against the reference gene elongation factor 1a (EF1a). The final value of relative quantification was described as the fold change of gene expression in the test sample compared to BSA-stimulated cells using −ΔΔCt analysis.

**RNA sequencing.** For whole-transcriptome analysis, human peripheral CD14^+^ monocytes were stimulated with BSA- or IgA-coated latex beads, after which RNA sequencing was performed. cDNA sequencing libraries were prepared with 1 μg of RNA input according to Illumina TruSeq Stranded mRNA Sample Preparation Guide (Illumina Inc., San Diego, CA, USA). In brief, total RNA was treated with DNase I and poly-T oligo attached magnetic beads to elute poly-(A+) mRNA. To synthesize the first-strand cDNA, purified mRNA was fragmented and primed using random-hexamer primers and reverse transcriptase. Second-strand cDNA was synthesized by incorporating dUTPs using DNA polymerase I. Subsequently, the 3′ ends of double-stranded cDNA were adenylated, and unique barcoding adapters were ligated to the ends. Barcode-ligated cDNA fragments were selectively enriched using PCR to create cDNA libraries for sequencing. Eighteen unique barcode sequences were applied for simultaneous analysis of multiple samples. The quality and yield of the resulting libraries were assessed using the Agilent D5000 ScreenTape (Agilent Technologies, Santa Clara, CA, USA) before sequencing. Libraries were loaded onto an Illumina cluster station (Illumina Inc., San Diego, CA, USA) and sequenced using Illumina HiSeq 4000 (Illumina Inc., San Diego, CA, USA). The optimal read depth to analyze the mRNA transcriptome of primary human monocytes was determined at 10 million reads per sample using single reads with 50 cycles per read.

**RNA sequencing data analysis.** Data were analyzed using the R-package ShrinkBayes [79,80], a software package dedicated to the analysis of RNAseq data. It uses the zero-inflated negative binomial distribution to model the over-dispersed count data and, unlike many other packages, accounts for a potential excess of zeros. Moreover, it applies shrinkage (i.e., borrowing information across genes) to provide stable estimates of differential expression effects and their uncertainties. This is particularly beneficial for small sample sizes. For two-group comparisons, the effect sizes are modeled by a spike-and-slab mixture with a spike on zero to capture the nondifferential genes, along with two Gaussian slabs to capture positive and negative differential effects. ShrinkBayes then computes the (posterior) probability that a gene is differentially expressed to provide a local false discovery rate (lfdr) estimate for each gene. For multigroup comparisons, a mixture of a spike on zero (no difference between groups) and noninformative Gaussian slabs was used to model the group-specific effect sizes. The Bayes factor, quantifying the relative likelihood of the absence of any group effect versus its presence, weighted by the prior odds of being non-differentially expressed, renders the lfdr [81]. These odds were estimated from the data using empirical Bayes [80]. It has been shown that ShrinkBayes results align well with simple false discovery rate (FDR)-adjusted Wilcoxon test-based p-values for large sample sizes, while being much more powerful for small ones [82]. Genes with lfdr ≤ 0.05 were considered significant.

**Luminex assay.** Cytokine and chemokine release of human peripheral CD14^+^ monocytes after 24 h of stimulation with BSA- or IgA-coated latex beads (E:T ratio 1:100) was measured with a Human Custom ProcartaPlex Multiplex 14-plex assay (PPX-08-MX9HJ3U, ThermoFisher, Waltham, MA, USA) using a Bio-Plex 200 (Bio-Rad, Hercules, CA, USA) according to the manufacturer’s instructions. The following markers were determined in cell culture supernatants: chemokine ligand 13 (CXCL13), chemokine ligand 20 (CCL20), interleukin-4 (IL4), interleukin-6 (IL6), interleukin-7 (IL7), interleukin-10 (IL10), and tumor necrosis factor-α (TNFα).

**Microscopy.** Colon biopsies were embedded using OCT compound (Sakura Finetek, Alphen aan den Rijn, The Netherlands), snap-frozen in liquid nitrogen, and kept at −80 °C. Tissue sections (5 μm) were fixed with acetone (10 min, RT), air-dried, and blocked with PBST with 2% BSA (30 min RT). Mouse αCD66b Alexa Fluor 488 (1:100, NovusBio cat: NB100-77808AF488, Littleton, CO, USA), mouse αCD19 Alexa Fluor 647 (1:50 SONY cat: 122388, Tokyo, Japan), and mouse αCD14 Alexa Fluor 488 (1:100 BD Biosciences cat: 555396, San Jose, CA, USA) were incubated in separate panels (1 h, RT). Slides were washed, treated with 1:1000 DAPI for 5 min, and embedded with Mowiol. Cryosections were analyzed using a Leica DM6000 microscope (Leica Microsystems B.V., Wetzlar, Germany). To quantify the infiltration of intestinal immune subsets, we calculated the surface ratio of CD66b, CD14, and CD19 within 10,000 µm^2^ of total tissue surface, thereby correcting for different sizes of biopsies using ImageJ software.

**Flow cytometry.** B cells from cocultures and single-cell cultures were harvested after 7 days and blocked in 2% BSA PBS (30 min, 4 °C). Cells were stained with goat αIgA F(ab)2 biotin (1:100 ThermoFisher cat: A24462, Waltham, MA, USA) in 0.1% BSA PBS (30 min, 4 °C). Cells were washed and incubated with streptavidin Alexa Fluor 488 (1:200 Molecular Probes cat: S11223, Eugene, OR, USA) and fixable cell viability dye ef780 (1:1000 eBioscience cat: 65-0865-14, Waltham, MA, USA) (30 min, 4 °C). After washing, cells were measured the same day using an LSR-Fortessa X20 (BD Bioscience, San Jose, CA, USA).

**Human IgA ELISA.** Supernatants were centrifuged at 1500 rpm twice to remove cells, after which supernatants were centrifuged twice at 14,000 rpm and filtered with a 0.22 µm multiScreen-GV filter plate (Merck, Kenilworth, NJ, USA) to remove beads. Flat-well microtiter ELISA plates (Nunc-Immuno MaxiSorp, ThermoFisher cat:44-2404-21, Waltham, MA, USA) were coated with αHu-IgA (1:250 BD Biosciences cat: 555886, San Jose, CA, USA), followed by a blocking step for nonspecific binding sites with PBST containing 0.5% BSA for 1 h at 37 °C. Coated plates were incubated with undiluted supernatants or different concentrations of pooled human serum IgA as a reference (1 h 37 °C). Plates were washed with PBST, followed by incubation with biotin-labeled mouse anti-human IgA mAbs (1:250 BD Biosciences cat: 555884, San Jose, CA, USA) (1 h 37 °C). Plates were washed and further incubated with streptavidin horseradish peroxidase (HRP) (30 min, RT). As substrate, (3,3′, 5,5′)-tetramethylbenzidine (TMB) was used. Plates were read using a microplate reader (Bio-Rad, Hercules, CA, USA) at 450 nm.

**Statistics.** Data analysis was performed using GraphPad Prism version 4.03 for Windows (GraphPad Software, San Diego, CA, USA). Data are expressed as the mean ± SD. Before performing statistical testing, it was determined whether data were normally distributed. If normally distributed, statistical differences were determined using two-tailed unpaired Student’s *t*-tests (comparing two groups) or paired *t*-test for matched patient sample calculation. If not normally distributed, a Wilcoxon matched rank test was applied between two matched groups. Differences were considered statistically significant for *p* < 0.05.

## Figures and Tables

**Figure 1 ijms-23-11132-f001:**
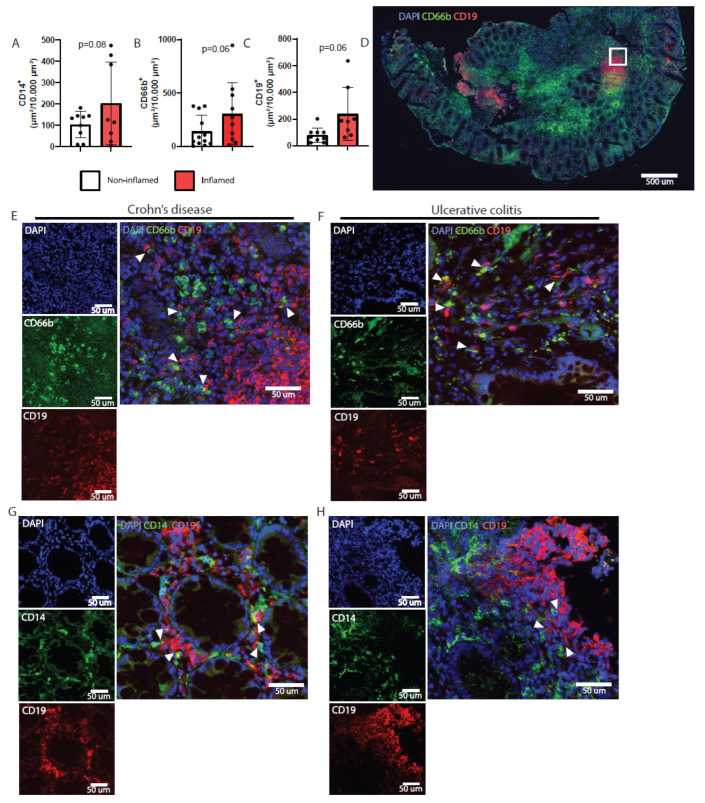
Myeloid cells interact with B lymphocytes in the inflamed colon tissue of inflammatory bowel disease patients. Tissue was collected from either inflamed pathological regions or adjacent macroscopic nonpathological parts of the colon of inflammatory bowel disease (IBD) patients. (**A**) Quantification of the number of monocytes (identified as CD14^+^), (**B**) neutrophils (identified by anti-CD66^+^), and (**C**) B lymphocytes (identified as CD19^+^) in IBD patient biopsies from inflamed (red) or noninflamed (white) regions of the colon. (**D**) Example image for DNA (DAPI, blue), CD66b^+^ cells (green), and B lymphocytes (CD19, red) in inflamed colon biopsy of Crohn’s disease patient. Magnification (white square) was used for (**E**). (**E**,**F**) Single staining of DNA, CD66b^+^ neutrophils, and CD19^+^ cells of (**E**) Crohn’s disease (representative sample out of *n* = 5) and (**F**) ulcerative colitis (representative sample out of *n* = 3). Merged channels visualize the interaction (white arrows) between CD66b^+^ cells and CD19^+^ B lymphocytes. (**G**,**H**) Visualization for DNA, CD14^+^ monocytes (green), and CD19^+^ B lymphocytes in inflamed colon biopsy of patients with (**G**) Crohn’s disease (*n* = 5) or (**H**) ulcerative colitis (*n* = 3). Merged channels are demonstrated to visualize the interaction (white arrows) between CD14^+^ cells and B lymphocytes.

**Figure 2 ijms-23-11132-f002:**
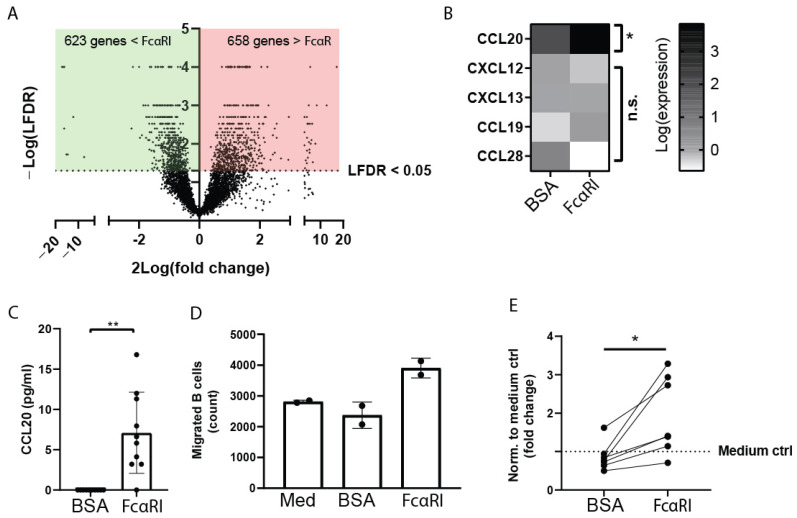
FcαRI-stimulated monocytes produce the chemokine CCL20 and attract B lymphocytes. (**A**,**B**) RNA sequencing of monocytes stimulated with either BSA- or IgA-coated (activating FcαRI) beads was performed. (**A**) Volcano plot of all genes expressed by monocytes with BSA- or IgA-coated bead stimulation. The *x*-axis represents log_2_ of the fold change, while the *y*-axis represents the local false discovery rate (lfdr). (**B**) Heat map of log-transformed RNA-seq counts of human peripheral monocytes after stimulation with BSA- or IgA-coated (activating FcαRI) beads. Genes above an lfdr of 0.05 are considered significant. Coloring represents log expression of expressed genes. (**C**–**E**) Peripheral monocytes were stimulated with BSA- or IgA-coated (activating FcαRI) beads for 24 h. Luminex was performed on the supernatant of stimulated monocytes to determine (**C**) C–C motif chemokine ligand 20 (CCL20). (**D**,**E**). Supernatants were placed in the bottom compartment of transwell chambers, while upper chambers contained peripheral B cells. Migration toward lower compartments was determined. (**D**) A representative experiment is shown (technical duplicate). Med: medium only as negative control. (**E**) Quantification of fold increase of B-cell migration toward supernatants of monocytes stimulated with BSA- or IgA-coated (activating FcαRI) beads. B-cell migration to medium was used to normalize for each donor was normalized. Data are presented as the mean ± SD. n.s. = not significant. Wilcoxon matched rank test, ** *p* < 0.01; Student’s *t*-test, * *p* < 0.05.

**Figure 3 ijms-23-11132-f003:**
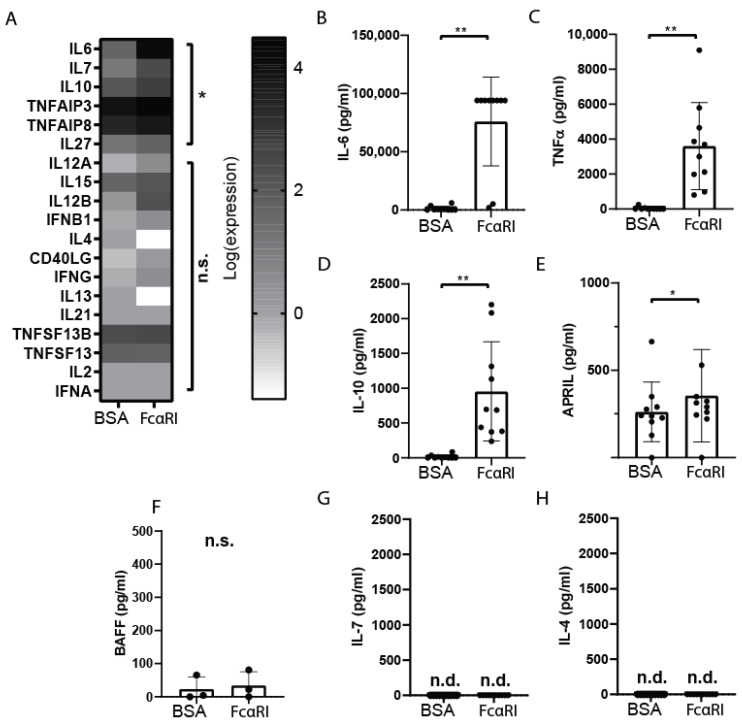
FcαRI-stimulated monocytes produce interleukin-6, interleukin-10, and tumor necrosis factor-α. Human peripheral monocytes were cultured with BSA- or IgA-coated (activating FcαRI) beads for 4 h. (**A**) Heat map of log-transformed RNA-seq counts of human peripheral monocytes stimulated with BSA- (*n* = 5) or IgA-coated (activating FcαRI) (*n* = 10) beads. Genes above an lfdr of 0.05 are considered significant. Coloring represents log expression of the expressed genes. Peripheral monocytes were cultured with BSA- (*n* = 10) or IgA-coated (activating FcαRI) (*n* = 10) beads (*n* = 10) for 24 h. Luminex was performed on supernatant of stimulated monocytes to determine (**B**) interleukin-6, (**C**) tumor necrosis factor-α (TNFα), (**D**) interleukin-10, (**E**) APRIL, (**G**) interleukin-7, and (**H**) interleukin-4 secretion. (**F**) Similarly, BAFF secretion was measured 24 h after BSA- or IgA-coated (activating FcαRI) bead stimulation (*n* = 3) using ELISA. Not detected protein (n.d.) is represented as zero. Data are presented as the mean ± SD. n.s. = not significant. Wilcoxon matched rank test, * *p* < 0.05, ** *p* < 0.01.

**Figure 4 ijms-23-11132-f004:**
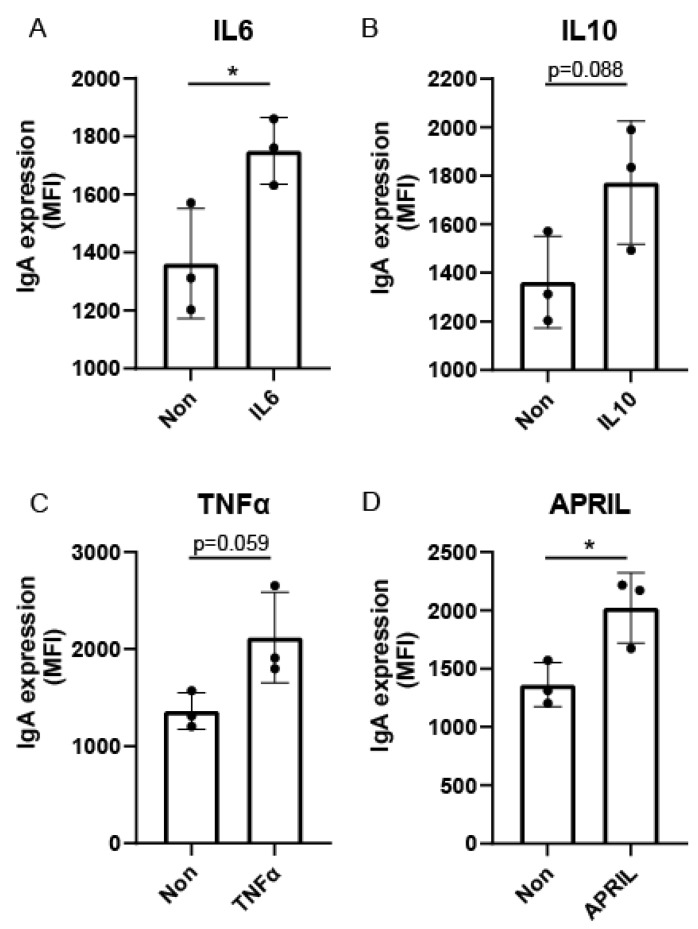
Exogenous recombinant cytokines induce IgA class switching in naïve B lymphocytes. Human naïve B cells were isolated from peripheral blood and cultured in the presence of T-cell-dependent stimuli, including interleukin-4, anti-CD40, and anti-IgM antibodies. In addition, B cells were supplemented with exogenous (**A**) interleukin-6, (**B**) interleukin-10, (**C**) tumor necrosis factor-α, or (**D**) APRIL. On day 7, IgA membrane expression was measured on B cells using flow cytometry (*n* = 3). Data are presented as the mean ± SD. Student’s *t* test, * *p* < 0.05.

**Figure 5 ijms-23-11132-f005:**
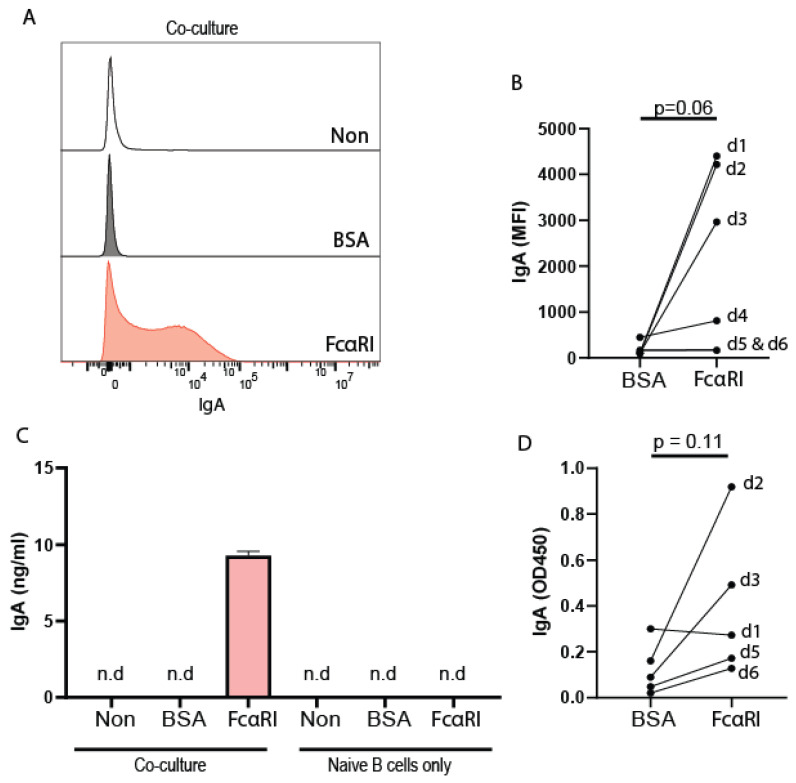
FcαRI-stimulated monocytes induce IgA class switching in naïve B lymphocytes. Monocytes were stimulated with BSA- or IgA-coated (activating FcαRI) beads and cocultured with peripheral naïve B cells for 7 days in the presence of interleukin-4, anti-IgM, and anti-CD40 antibodies. (**A**) Representative histogram demonstrating IgA surface expression by B lymphocytes after coculture with monocytes treated without beads (non; white), with BSA-coated beads (gray) or IgA-coated beads (FcαRI) (red). (**B**) Quantification of the mean fluorescent intensity of IgA on B lymphocytes of six donors. (**C**) Representative ELISA demonstrating secreted IgA in the supernatants of co-cultured naïve B cells with monocytes treated without beads (non), with BSA-coated beads, or with IgA-coated beads (red). Naïve B cells in the absence of monocytes were cultured with beads as a negative control. (**D**) Quantification of IgA secretion by B cells in five donors. Donor numbers in (**B**,**D**) correspond. Not enough supernatant of d4 was available to analyze IgA secretion. Data are presented as the mean ± SD.

**Figure 6 ijms-23-11132-f006:**
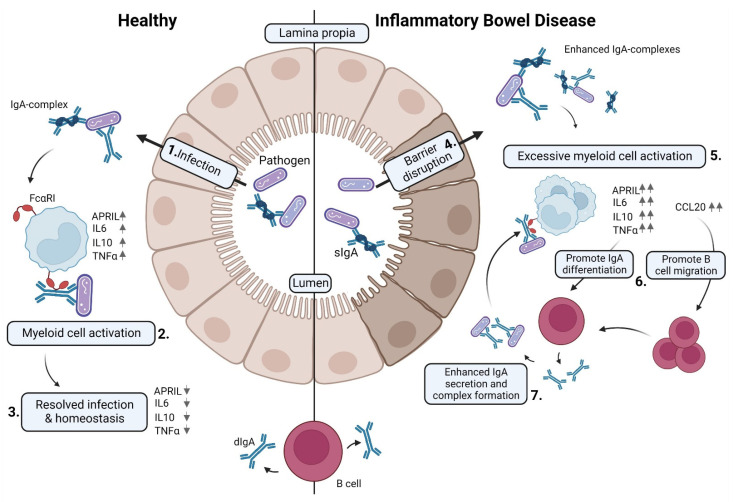
FcαRI-activated monocytes influence B lymphocyte migration and differentiation. **Left panel**: (1) During a mucosal infection, dIgA-opsonized bacteria are recognized by myeloid cells and cleared (2). (3) B-cell activation factors are temporarily increased; however, once the infection is cleared the process stops, and the system returns to homeostasis. **Right panel**: IBD is characterized by (4) a disrupted epithelial lining. (5) Excessive IgA complexes activate myeloid cells via FcαRI, which leads to production of cytokines that (6) promote B-cell migration to the site of inflammation via CCL20 and skewing into an IgA isotype. (7) In turn, this mechanism may promote enhanced IgA production and concomitant IgA complexes, thereby continuously stimulating newly recruited myeloid cells. As such, a perpetuating positive feedback-loop may be initiated that activates myeloid cells and promotes IgA B-cell differentiation.

## Data Availability

Not applicable.

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
