# Peer review of "Fcα Receptor-1-Activated Monocytes Promote B Lymphocyte Migration and IgA Isotype Switching"

_ijms, 2022, doi:10.3390/ijms231911132_

Round 1

Reviewer 1 Report

In this study, the authors explored the role of FcαRI-stimulated myeloid cells on B lymphocyte migration and IgA isotype switching, which may play a key role in inflammatory bowel disease (IBD) pathology. Blocking FcαRI with monoclonal antibodies can reduce IgA production and may become a novel therapeutic intervention for IBD treatment. This study is very interesting, and the manuscript is very well written. It could be relevant to the field. However, there are few concerns and my specific comments are:

1.     Authors need to correct typographical errors in the manuscript. For example in  the abstract, line No: 5

2.     In the figure legend (Figure 6), author should check the font style

3.     Authors should use full form of the abbreviations for the first time. For example BSA and PSG

4.     What is the role of neutrophils in IgA induction in this study? In literature, it has reported that the neutrophils negatively regulate the stimulation of mucosal IgA responses in sublingual immunization. 

Author Response

  1. Authors need to correct typographical errors in the manuscript. For example in  the abstract, line No: 5

Response: Thank you for pointing out these irregularities. We have re-read the manuscript and corrected for typographical errors. Adjustments were made using Word track-and-trace.

  1. In the figure legend (Figure 6), author should check the font style

Response: We have used bolt font styles to indicate left and right panel, for emphasis, and would like to keep this for clarity. We changed the semicolon into a colon.

  1. Authors should use full form of the abbreviations for the first time. For example BSA and PSG.

Response: Abbreviations were written in full form (incl BSA, PSG). Adjustments were made using Word track-and-trace.

  1. What is the role of neutrophils in IgA induction in this study? In literature, it has reported that the neutrophils negatively regulate the stimulation of mucosal IgA responses in sublingual immunization. 

Response: The  manuscript to which the reviewer refers describes the role of murine neutrophils on suppressing antigen-specific serum/fecal IgA responses. It was demonstrated that sublingual immunization in mice resulted in enhanced specific IgA secretion when neutrophils had been depleted, suggesting that neutrophils suppress the production of IgA upon immunization.

However, mice studies cannot be directly compared with human studies, because of a notable difference i.e. the absence of FcαRI in mice.

In our study we propose that during IBD pathogenesis myeloid cells become activated via FcαRI, which results in cytokine/chemokine production that leads to enhanced B cell IgA differentiation. We have mostly used monocytes to perform functional in-vitro assays to support this hypothesis as neutrophils are practically difficult to work with due to their short half-life. Long-term in vitro co-cultures with B lymphocytes resulted in a dramatic loss of B cell viability, presumably because of the presence of dead neutrophils (and their proteases) (Supplemental figure 5). However, we and others have published that crosslinking FcαRI on human neutrophils triggers potent pro-inflammatory functions including degranulation, phagocytosis, chemotaxis, antibody-dependent cellular cytotoxicity [references 27-29 in paper] and production of pro-inflammatory cytokines including IL6, TNFα and IL1β [reference 25]. This will not occur with murine neutrophils, as they don’t express FcαRI.

We have included this issue into the discussion (page 11, and 53 https://www.ncbi.nlm.nih.gov/pmc/articles/PMC4481173/)

Whereas  murine neutrophils may suppress mucosal IgA production, this model excludes the role of FcαRI, and as such is not comparable to the human situation.

Reviewer 2 Report

In this manuscript, Bos et al. presented that FcαRI-activated monocytes produced a cocktail of cytokines and chemokines that stimulated IgA switching in B cells, and close contact between B cells and myeloid cells was observed in the colons of IBD patients. This is an innovative topic as little is known. Nevertheless, there are significant drawbacks to the manuscript and there are some issues that raise concerns and they should be addressed, especially there is no difference in most of the data. In summary, the premise is not refined, there are many missing methodological details, some important drawbacks to the study design, and an incomplete discussion of study limitation. 

1.    In the figure 1A-C, the quantification of the number of monocytes, neutrophils and B lymphocytes just have an increased trend in IBD patients, while it is no significant change compared to paired non-inflamed tissue. The authors must increase the sample to confirm that B lymphocytes and myeloid cells are significantly increased to determine that they are indeed involved in IBD pathogenesis, otherwise no significant difference is equivalent to no change.

2.     The same problem occurs in Figures 4B, C and some supplementary data. Authors can increase the number of repetitions to see if it has significantly increased.

3.    If the author think that high neutrophil infiltration leads to more interaction with B lymphocytes. Figure 1E-H should show more interactions in the Crohn’s disease. The author needs to be quantified to support the statement. A single high-powered photomicrograph is not sufficient.

4.    The author explain the FcαRI-stimulated monocytes induce IgA class switching is not clearly. 

Minor Comments:

1.     The histogram of Figure 2D does not label the “N” value.

2.     The vertical axis in the bar chart of Figure 3E is not fully label. Figure 4A, B shows incomplete.

Author Response

  1. In the figure 1A-C, the quantification of the number of monocytes, neutrophils and B lymphocytes just have an increased trend in IBD patients, while it is no significant change compared to paired non-inflamed tissue. The authors must increase the sample to confirm that B lymphocytes and myeloid cells are significantly increased to determine that they are indeed involved in IBD pathogenesis, otherwise no significant difference is equivalent to no change.

Response:

We agree with the reviewer that increasing the number of samples would be preferential. Unfortunately, we only have a limited amount of tissue biopsies available and could maximally increase our cohort with three extra paired samples for the CD66b+ neutrophil staining and one for the CD14+ monocyte staining. We are regrettably not in the position to expand the amount of colon biopsies on short notice, as the permit from our medical ethics committee has expired, and there is a huge backlog to obtain permission for non-COVID-19 related studies. After inclusion of the extra samples the p values chanced into p=0.08 (monocytes), and p=0.06 (CD66+ neutrophils). Although still not significantly different, our results do suggest a trend towards increased infiltration in inflamed tissue of IBD patients, which is in line with multiple studies described in literature  (e.g. monocytes: Bernardo et al., 2018; Chapuy et al., 2019; Ogino et al., 2013; Thiesen et al., 2014) and neutrophils (van der Steen et al., 2009)). We therefore suggest a trend towards increase and included extra references to support this point.

  1. The same problem occurs in Figures 4B, C and some supplementary data. Authors can increase the number of repetitions to see if it has significantly increased.

Response: We are currently performing additional experiments to increase the amount of replicates for figure 4.

  1. If the author think that high neutrophil infiltration leads to more interaction with B lymphocytes. Figure 1E-H should show more interactions in the Crohn’s disease. The author needs to be quantified to support the statement. A single high-powered photomicrograph is not sufficient.

Response: We aimed to suggest that the chance for interaction between myeloid cells and B cells is likely increased if more cells are present in inflamed tissue. However, as we do not demonstrate quantifications to further support this statement, we agree with the reviewer that this statement is too strong. We have adjusted the manuscript.

  1. The author explain the FcαRI-stimulated monocytes induce IgA class switching is not clearly. 

Response: To better explain the proposed mechanism that FcαRI-stimulated myeloid cells may contribute to IgA B cell differentiation, we have further elaborated in summary figure 6. We have included numbers to visualize that during IBD pathogenesis, we anticipate that excessive IgA complexes activate myeloid cells via FcαRI leading to production of cytokines and chemokines, which  promote B cell migration to the site of inflammation and skewing into an IgA isotype. In turn, this may promote enhanced IgA production and concomitant IgA complexes, hereby continuously stimulating newly recruited myeloid cells. As such, a perpetuating positive feedback-loop may be initiated that activates myeloid cells and promotes IgA B cell differentiation.

Minor Comments:

  1. The histogram of Figure 2D does not label the “N” value.

Response: We have included the individual data points for the representative duplicate, as suggested by the reviewer.

  1. The vertical axis in the bar chart of Figure 3E is not fully label. Figure 4A, B shows incomplete.

Response: We think the figures may have been incomplete due to a conversion error in our pdf file, as in our Word file we cannot detect the described errors observed by the reviewer. We have rearranged the positions of the figures and hope that this allows appropriate representation of our figures.

Round 2

Reviewer 2 Report

The revised version of manuscript was significant improved the quality and address my questions. I believe the revised manuscript meet the sufficient warrant publication in journal.

Author Response

The reviewer has no further questions, and we are pleased to hear the manuscript is ready for publication.